# DiffER²: Diffusion Ensembles for Retrosynthesis Prediction with SMILES Adapted Particle Guidance

## Abstract

Computational retrosynthesis prediction has the potential to reduce development time for newly discovered drugs by automatically generating potential reactions for a target product. Data-driven approaches commonly treat this as a sequence-to-sequence generation task on SMILES strings. In this work, we construct ensembles of discrete-time and continuous-time diffusion models for molecular generation and incorporate guidance mechanisms for improved output diversity. We propose an adaptation of particle guidance to SMILES sequence generation which significantly improves the number of unique molecules generated by diffusion ensembles while increasing top-k accuracy. These results further expand the efficacy of discrete diffusion for SMILES generation, and our empirical analyses offer new insights into the capabilities of diffusion models for chemical retrosynthesis.

## 1 Introduction

The goal of retrosynthesis is to construct reactant molecules that can be used to synthesize a target product. Retrosynthesis prediction constitutes a pivotal step in the drug manufacturing pipeline. Repeatedly applying single-step retrosynthesis models to a given problem can result in full synthetic pathway discovery (Segler et al., 2018; Shen et al., 2021), further supporting pharmaceutical advancement for new drug modalities. However, even single-step retrosynthesis prediction is difficult, in part due to the diversity of reactant sets that might map to a single product. While forward synthesis prediction is primarily considered a surjective process with a deterministic output, retrosynthesis prediction is in many ways the opposite, with many possible outputs resulting from a single input. As a result, researchers have taken a particular interest in deep learning methods that hasten the process of retrosynthesis discovery by quickly and efficiently predicting multiple possible reactant sets from a given product molecule.

Numerous methodologies have been developed, from template-based methods that use mined reaction mechanisms to predict reactants to template-free approaches that directly predict string or graph representations of reactant molecules conditioned on the target product. The most successful of these template-free approaches use traditional transformer encoder-decoder architectures on SMILES (Weininger, 1988) string representations of the molecules, effectively treating the problem as a machine translation task (Zhong et al., 2022). A recent approach, DiffER (Current et al., 2024), explored the usage of discrete diffusion models for the retrosynthesis task and provided evidence that an ensemble of discrete-time diffusion models could be a practical approach for sampling from the posterior distribution of reactants. Differ outperformed the state-of-the-art R-SMILES (Zhong et al., 2022) at top-1 accuracy on the commonly used USPTO-50k dataset (Lowe, 2017), but suffered from low output diversity due to the nature of posterior sampling.

In this work, we extend upon the work of DiffER (Current et al., 2024) by adapting their approach to include continuous-time discrete diffusion models and applying classifier-based particle guidance (Corso et al., 2023) to SMILES string generation. Both of these changes aim to improve the efficacy and output diversity of models over the original DiffER. A visual overview of the DiffER² ensemble approach is presented in Figure 1. The main improvements on the original DiffER methodology are as follows:

Figure 1: Overview of the DiffER$^2$ ensemble. The DiffER$^2$ ensemble contains both continuous-time (C-DiffER) and discrete-time (DiffER) models to leverage differing capabilities between the two diffusion archetypes. Both normal sampling methods and SMILES adapted particle guidance are used to generate output. Output SMILES are ranked according to frequency of the canonical form. DiffER$_\kappa$ indicates the value of $\kappa$ as described in Section 3.2.

1. We incorporate continuous-time discrete diffusion models into the DiffER ensemble to leverage different capabilities of discrete-time and continuous-time diffusion.

2. We explore the efficacy of SMILES-adapted guidance mechanisms to further enhance the diversity and accuracy of diffusion models for chemical retrosynthesis.

3. Finally, we offer an in-depth exploration into the differing performance capabilities of auto-regressive and discrete-space diffusion models on the chemical retrosynthesis task.

By combining both continuous-time and discrete-time diffusion models with classifier-based particle guidance (Corso et al., 2023), we significantly improve ensemble performance and reduce oversampling of posterior peaks while maintaining or boosting top-k accuracy across the board. Our results enhance the efficacy of diffusion ensembles for SMILES generation while providing an empirical analysis of diffusion performance compared to auto-regressive approaches for sequence generation.

## 2 RELATED WORK

### 2.1 RETROSYNTHESIS PREDICTION

Data-driven models for retrosynthesis prediction are commonly divided into three approaches: template-based, semi-template, and template-free methods. Common template-based approaches include GLN (Dai et al., 2019) and LocalRetro (Chen & Jung, 2021). These approaches primarily focus on identifying locations in a molecule where reaction templates can be applied and ranking the efficacy and feasibility of those templates on the target area. Semi-template methods include the likes of GraphRetro (Somnath et al., 2021), G$^2$Retro (Chen et al., 2023), and GDiffRetro (Sun et al., 2025). These approaches generally work in a two-step manner: in the first step, a reaction center is first predicted from the product molecule, and the molecule is subsequently split into synthons. In the second step, each synthon is completed to construct a final reactant molecule. Notably, GDiffRetro (Sun et al., 2025) uses discrete-time diffusion processes to complete predicted synthons. Finally, template-free methods such as Chemformer (Irwin et al., 2022), MEGAN (Sacha et al., 2021), R-Smiles (Zhong et al., 2022), and DiffER (Current et al., 2024) directly construct reactant molecules from the product molecule. This is most often done in a sequence-to-sequence manner using SMILES strings (Irwin et al., 2022; Zhong et al., 2022; Current et al., 2024), though some approaches directly operate on graphs, such as the case of MEGAN (Sacha et al., 2021), which iteratively builds the reactant molecule(s) through a series of edits. DiffER (Current et al., 2024) is the first approach to use conditional discrete diffusion to accomplish the retrosynthesis prediction task, although their methodology suffers from reduced output diversity and high computational cost compared to auto-regressive decoding strategies. We primarily focus on template-free methods for performance comparison.

## 2.2 DISCRETE-SPACE DIFFUSION

Generative diffusion models have been a significant topic of research in recent years, and have achieved significant success in the realms of image generation (Croitoru et al., 2023; Yang et al., 2023) and signal processing (Yang et al., 2023). Though diffusion models traditionally operate in the continuous regime (Sohl-Dickstein et al., 2015; Song & Ermon, 2019; Song et al., 2020; Ho et al., 2020; Saharia et al., 2022; Gu et al., 2022; Zhang et al., 2023), significant effort has been made extending diffusion models to discrete spaces (Hoogeboom et al., 2021; Austin et al., 2021; Gong et al., 2022; 2023; He et al., 2022; Yuan et al., 2022; Dieleman et al., 2022; Ghazvininejad et al., 2019). Discrete-space models have demonstrated remarkable performance for molecular generation (Xu et al., 2022; Alakhdar et al., 2024; Hoogeboom et al., 2022; Xu et al., 2023; Schneuing et al., 2024) and protein design (Gruver et al., 2023; Watson et al., 2022; Ni et al., 2023), but have struggled to out-compete their auto-regressive counterparts on discrete sequence generation. Furthermore, most work in this space has primarily focused on discrete-time diffusion. Only recently has discrete-space diffusion been extended to continuous-time implementations (Campbell et al., 2022; Zhao et al., 2024; Xu et al., 2024), which demonstrate improved modeling capabilities and sampling efficiency compared to their discrete-time parallels. Furthermore, incorporating recent advances in guidance mechanisms during diffusion sampling (Bansal et al., 2023; Liu et al., 2023; Epstein et al., 2023; Ho & Salimans, 2022) remains a challenge, due to the difficulty of applying gradient-based approaches on a discrete domain (Schiff et al., 2024; Gruver et al., 2023). Schiff et al. (2024) recently adapted classifier-free and classifier-based guidance mechanisms for discrete diffusion and demonstrated promising results on molecular generation tasks, though more work needs to be done constructing and evaluating guidance mechanisms for discrete diffusion tasks.

## 3 METHODS

Following the original DiffER (Current et al., 2024), we treat retrosynthesis prediction as a discrete sequence-to-sequence generation task, where the target product is a conditional input to the diffusion generation akin to text-to-image models. We employ both discrete-time and continuous-time diffusion models. Like DiffER, we incorporate length prediction strategies to improve diffusion sampling as well as enable robust model ensembling. Finally, we adapt classifier-based particle guidance (Corso et al., 2023) to SMILES sequence generation to improve output diversity and reduce over-sampling of posterior peaks. By applying our techniques to SMILES generation, we are able to offer direct comparisons to prior works, and more clearly demonstrate the effectiveness of our diffusion ensembling and guidance methodologies. However, we recognize the existence of other molecular string representations such as SELFIES (Krenn et al., 2019). We offer a comparison to a SEFLIES-based implementation of our work in Appendix A.4.

### 3.1 PRELIMINARIES

Following the notation of Zhao et al. (2024), let $x_0 \sim p_{data}(x_0)$ be a categorical random variable representing the observed data with discrete distribution $p_{data}(x_0; K)$. Let $x_t \sim q(x_t)$ be the latent variable at time $t \in [0, 1]$ (continuous-time, max time $T = 1$) or $t \in \{1, ..., T\}$ (discrete-time) and $x_{t|s} \sim q(x_t|x_s)$ be the conditional random variable. Let $x_t^{1:L_x} \in \{0, 1\}^{L_x \times K}$ be a sequence of length $L_x$ at time $t$, and let $x_t^{[\ell]}$ be the sequence indexed at position $\ell$. We assume each discrete random variable is governed by a categorical distribution $\mathcal{C}$ with probability distribution $p \in [0, 1]^K$, where $||p||_1 = 1$, such that $x_t \sim \mathcal{C}(x_t; p)$. Furthermore, let $Q_{t|s} \in [0, 1]^{K \times K}$ represent the transition matrix between times $s$ and $t$, such that $[Q_{t|s}]_{ij} = q(x_t = e_i|x_s = e_j)$, where $e_i, e_j \in \{0, 1\}^K$ are one-hot encoded vectors.

In the forward diffusion process, noise is added to observed data until it is indistinguishable from a stationary noise distribution. Zhao et al. (2024) show that the forward diffusion process for discrete-time and continuous-time diffusion can be unified into a single forward diffusion process defined by the accumulated transition matrix $\bar{Q}_{t|s}$, with $t > s$:

$$\bar{Q}_{t|s} = \bar{\alpha}_{t|s}I + (1 - \bar{\alpha}_{t|s})1m^\top, \tag{1}$$

where $m$ is a known probability distribution vector such that $x_T \sim \mathcal{C}(x_T; m)$, 1 is vector of 1s, and $\bar{\alpha}_{t|s}$ is the accumulated noise schedule between time $t$ and time $s$.

In the backward diffusion process, diffusion models aim to generate samples from $p_\theta(x_s|x_t)$ for some $0 < s < t \leq T$. This is commonly accomplished by taking $p_\theta(x_s|x_t) \approx q(x_s|x_t, x_0)$, where $q(x_s|x_t, x_0)$ is the $s$-step posterior distribution (Hoogeboom et al., 2021; Austin et al., 2021). Because $x_0$ is unknown, it is approximated by training a neural network model $f_t^\theta$ such that $p_\theta(x_0|x_t) = \mathcal{C}(x_0; f_t^\theta(x_t))$. Thus, $p_\theta(x_s|x_t)$ can be estimated by sampling $x_s \sim q\left(x_s|x_t, f_t^\theta(x_t)\right)$. Analytically, this can be calculated as

$$q\left(x_s|x_t, f_t^\theta(x_t)\right) = \mathcal{C}\left(x_s; \frac{\bar{Q}_{t|s} x_t \odot \bar{Q}_s^\top f_t^\theta(x_t)}{x_t^\top \bar{Q}_t^\top f_t^\theta(x_t)}\right). \tag{2}$$

This posterior calculation can be further reduced to simplify computation (Zhao et al., 2024), which we follow in our work as well. We refer readers to Zhao et al. (2024) for a more detailed overview of the unified diffusion methodology.

The primary point of difference between discrete-time and continuous-time diffusion models now lies in the loss functions. We train both continuous-time and discrete-time diffusion models using the simplified variational lower bound loss functions provided by Zhao et al. (2024) alongside a cross-entropy loss.

## 3.2 DiffER Adapted Unified Diffusion

DiffER (Current et al., 2024) incorporates the target product as an additional constraint to the diffusion generation process, akin to text-to-image generative diffusion models. Let $y_0^{1:L_y} \in \{0, 1\}^{L_y \times K}$ be the target product sequence associated with the reactant sequence $x_0^{1:L_x}$. Here, both $y_0$ and $x_0$ are SMILES strings representing molecular structures, with $x_0$ containing one or more concatenated reactant SMILES. Because the target product is known in retrosynthesis prediction, The diffusion problem can be parameterized such that

$$p_\theta(x_0|x_t, y_0) = \mathcal{C}(x_0; f_t^\theta(x_t, y_0)). \tag{3}$$

This formulation guides the diffusion process by incorporating the target product as an additional input to the model. Here, $f_t^\theta$ usually takes the form of an encoder-decoder model, where the encoder processes $y_0$ and the decoder acts as the diffusion component on $x_t$.

Notably, one issue remains in that the length of the reactant sequence $L_x$ is unknown a priori, which is necessary to properly instantiate $x_T \sim \mathcal{C}(x_T; m)$. Inspired by MaskPredict (Ghazvininejad et al., 2019), DiffER (Current et al., 2024) incorporates a length prediction component into the encoder half of $f_t^\theta$, such that $L_x$ can be estimated using some model $f_\ell^\phi : \{0, 1\}^{L \times K} \to \mathbb{R}^L$, so that $L_x \approx \arg\max f_\ell^\phi(y_0)$, where, $L$ is an arbitrary maximum sequence length such that $L_x < L$ and $L_y < L$ for all $x_0, y_0$ in the data distribution. For the chemical retrosynthesis task, $L_x$ and $L_y$ are often highly correlated: thus, to encourage active learning, $f_\ell^\phi$ is instead trained to predict the difference between $L_x$ and $L_y$, i.e.,

$$L_x = L_y + (\arg\max f_\ell^\phi(y_0) - \Gamma), \tag{4}$$

where $\Gamma < L$ is a sufficiently large constant allowing $f_\ell^\phi$ to effectively predict negative length changes. In practice, we set $\Gamma = \frac{L}{2}$, noting that the change in length will never be larger than half of the product sequence.

Current et al. (2024) show experimentally that if $f_\ell^\phi$ is a perfect length-prediction model, then discrete-time diffusion outperforms all other methodologies for top-1 predictions by a significant margin, exhibiting that sequence length information can be highly informative when predicting reactant SMILES. However, the authors note that in practice, $f_\ell^\phi$ cannot be sufficiently trained to accomplish such a notable performance increase due to 1.) the distribution of $(L_x - L_y)$ is heavily biased, 2.) the SMILES encoding of a molecule can produce sequences of varying length, forcing $f_\ell^\phi(y_0)$ to learn a one-to-many mapping, and 3.) accurate length prediction requires complex understanding of chemical reagents, molecular reactivity, and molecular structure, and is itself a nontrivial problem.

To remedy problems 1. and 2., DiffER (Current et al., 2024) randomly appends $L_p \sim \mathcal{U}\{1, \kappa\}$ padding tokens to the SMILES representations of reactants, where $\kappa$ is a set hyperparameter. This

padding procedure smooths the distribution of $(L_x - L_y)$ and allows flexibility in the output size of generated reactants. A detailed overview of this procedure is provided in Appendix A.1. To reduce reliance on the choice of $\kappa$, DiffER trains multiple diffusion models $f_t^{\theta_\kappa}$ with different $\kappa$, producing an ensemble of models which generate possible reactants. This approach provides better coverage of the posterior distribution than a single model. We follow this same approach, but additionally incorporate continuous-time models and classifier-based guidance sampling into the ensemble.

## 3.3 SMILES Adapted Particle Guidance for Chemical Generation

Aside from the incorporation of continuous-time diffusion models into the DiffER methodology and the formalization of said methodology, our other primary motivation of this work is to explore the usage of classifier-based guidance mechanisms on the DiffER ensemble models to improve output diversity. Classifier-based guidance steers diffusion sampling toward a specific classification $c$ by sampling from a tempered distribution $p^\gamma(x_s|c, x_t) \propto p(c|x_s)^\gamma p_\theta(x_s|x_t)$, where $p_\theta(x_s|x_t)$ is the diffusion model, $p(c|x_s)$ is a classifier, and $\gamma$ is the temperature parameter (Schiff et al., 2024):

$$\log p^\gamma(x_s|c, x_t) = \gamma \log p(c|x_s, x_t) + \log p(x_s|x_t). \tag{5}$$

Schiff et al. (2024) formalize classifier-based guidance for discrete-space diffusion specifically, with the assumption that $p^\gamma(x_s|c, x_t)$ factorizes independently across tokens. We follow their work to apply classifier-based guidance to our models.

### 3.3.1 SMILES Adapted Particle Guidance

One of the primary limitations of employing diffusion models for highly conditioned molecular generation tasks lies in the over-sampling of posterior peaks (Current et al., 2024; Sun et al., 2025), resulting in reduced diversity and number of unique generated molecules. To this end, we take inspiration from Corso et al. (2023) to encourage greater output diversity during sampling through guidance. Corso et al. apply the gradient of a potential function $\log \Phi_t(x_t^{(1)}, x_t^{(2)}, ..., x_t^{(n)})$ to encourage greater output diversity, where $x_t^{(i)}$ are independent samples from the diffusion model at time $t$. The potential function is constructed such that higher scores are given to distant point pairs:

$$\log \Phi_t(x_t^{(1)}, x_t^{(2)}, ..., x_t^{(n)}) = \frac{\bar{\alpha}_t}{2} \sum_{i,j} \mathcal{K}(x_t^{(i)}, x_t^{(j)}), \tag{6}$$

where $\mathcal{K}$ is a pairwise similarity kernel. Furthermore, the potential function is weighted by the noise schedule $\bar{\alpha}_t$ such that the guidance effect is stronger at earlier timesteps rather than later ones. Unfortunately, it is non-trivial to construct a differentiable similarity kernel between noised SMILES strings due to the non-singular nature of SMILES representations: the same molecule may be represented by many different SMILES sequences of various atom orderings.

To remedy this, we construct the potential function such that it maximizes the probability that $X_t = \{x_t^{(1)}, x_t^{(2)}, ..., x_t^{(n)}\}$ are noisy variants of different molecules,. Let $m_0^{(i)}$ be the molecular form of the denoised sequence $x_0^{(i)}$. Then we aim to maximize the probability

$$\bigcup_{i \neq j} p(m_0^{(i)} \neq m_0^{(j)}|x_t^{(i)}, x_t^{(j)}) = \prod_{i \neq j} p(m_0^{(i)} \neq m_0^{(j)}|x_t^{(i)}, x_t^{(j)}), \tag{7}$$

Which factors as a result of the independence of generated SMILES. Taking the $\log$, we can construct the potential function as

$$\log \Phi_t(X_t) = \sum_{i \neq j} \log p(m_0^{(i)} \neq m_0^{(j)}|x_t^{(i)}, x_t^{(j)}). \tag{8}$$

We train a classifier to estimate $p(m_0^{(i)} \neq m_0^{(j)}|x_t^{(i)}, x_t^{(j)})$. Recognizing that the parameterized diffusion model $f_t^\theta$ is already trained to produce an estimate of the de-noised sequences, we train an additional classifier $f_{\mathcal{K}}^\Phi$ to detect if $\hat{x}_{0,t}^{(i)} = f_t^\theta(x_t^{(i)})$ and $\hat{x}_{0,t}^{(j)} = f_t^\theta(x_t^{(j)})$ represent the same denoised molecular structure, i.e.,

$$p(m_0^{(i)} \neq m_0^{(j)}|x_t^{(i)}, x_t^{(j)}) = f_{\mathcal{K}}^\Phi(\hat{x}_{0,t}^{(i)}, \hat{x}_{0,t}^{(j)}) = (f_{\mathcal{K}}^\Phi \circ f_t^\theta)(x_t^{(i)}, x_t^{(j)}). \tag{9}$$

This parameterization results in the final potential function

$$\log \Phi_t(X_t) = \sum_{i \neq j} \log \left( (f_{\mathcal{K}}^\Phi \circ f_t^\theta)(x_t^{(i)}, x_t^{(j)}) \right). \tag{10}$$

### 3.3.2 APPLYING POTENTIAL FUNCTIONS VIA CLASSIFIER-BASED GUIDANCE

Rather than using this potential function directly, we instead construct a classifier on $x_t^{(i)}$ alone, recognizing that applying the classifier individually for all $x_t^{(i)} \in X_t$ reproduces the potential function:

$$\log p_\Phi(\mathbb{I}[\hat{m}_{0,t}^{(i)} \notin \hat{M}_t/\{\hat{m}_{0,t}^{(i)}\}]|x_t^{(i)}, X_t) = \sum_{j \neq i} \log \left( (f_\mathcal{K}^\Phi \circ f_t^\theta)(x_t^{(i)}, x_t^{(j)}) \right). \tag{11}$$

Here, $\hat{m}_{0,t}^{(i)}$ is the molecular form of $\hat{x}_{0,t}^{(i)}$, $\hat{M}_t$ is the set of molecules $\{\hat{m}_{0,t}^{(1)}, ..., \hat{m}_{0,t}^{(n)}\}$ associated with $X_t$, and $\mathbb{I}$ is the indicator function. For the sake of clarity, we will refer to $\mathbb{I}[\hat{m}_{0,t}^{(i)} \notin \hat{M}_t/\{\hat{m}_{0,t}^{(i)}\}]$ as the guidance class $c$. We add the further stipulation that if $\hat{x}_{0,t}^{(i)}$ is not a valid SMILES string or if $\hat{m}_{0,t}^{(i)}$ is not a valid molecule, then the indicator function defaults to False.

We finalize our guidance strategy by follow the approach of Schiff et al. (2024) to apply classifier guidance to discrete sequences. This results in the posterior estimate

$$\log p_{\Phi,\theta}^\gamma(x_s|c, x_t) = \sum_{\ell=1}^{L} \log \left( \frac{p_\Phi(c|\tilde{x})^\gamma p_\theta(x_s^{[\ell]}|x_t)}{\sum_{\tilde{x}} p_\Phi(c|\tilde{x})^\gamma p_\theta(x_s^{[\ell]}|x_t)} \right), \tag{12}$$

where $\tilde{x} = [x_t^{1:\ell-1}, x_s^\ell, x_t^{\ell+1:L}]$ is the sequence such that $\tilde{x}$ and $x_t$ are equal at all positions $\ell' \neq \ell$ and $\tilde{x}$ and $x_s$ are equal at position $\ell$. We set $\gamma = \bar{\alpha}_t$ to weight the guidance effect similarly to Equation 6. Finally, we use the first-order Taylor approximation of $\log p_\Phi$ to efficiently compute the classifier $p_\Phi(c|\tilde{x})$ using only a single forward pass (Schiff et al., 2024):

$$\log p_\Phi(c|\tilde{x}) = (\tilde{x} - x_t)^\top \nabla_{x_t} \log p_\Phi(c|x_t) + \log p_\Phi(c|x_t). \tag{13}$$

We refer readers to Schiff et al. (2024) for detailed proofs of this result.

Training and implementation details for $f_\mathcal{K}^\Phi$ are provided in Appendix A.2. During inference, $f_\mathcal{K}^\Phi$ is incorporated in a manner reminiscent of reinforcement learning, allowing the model to learn and adapt throughout the diffusion process.

## 4 EXPERIMENTS

**Experimental Setup.** We train and test our models on the USPTO-50K dataset (Lowe, 2017), a commonly used set of patented reactions that provides SMILES strings for both reactants and products and widely regarded as the benchmark dataset for retrosynthesis prediction. Reaction solvents are not included in the retrosynthesis task. Each pair of products and reactants is augmented 20 times according to Zhong et al. (2022), producing a final dataset of 1M root-aligned SMILES pairs. We use the augmented train, validation, and test sets provided by Zhong et al.[1]

The DiffER² ensemble contains eight individually trained diffusion models, depicted in Figure 1. Each model shares the same network architecture, differentiated only by the diffusion parameters and padding augmentation described in Section 3.2. Four of the diffusion models follow the original DiffER training (Current et al., 2024) with losses adjusted according to Zhao et al. (2024): these models operate in discrete-time with padding parameter $\kappa \in \{20, 40, 60, 80\}$. The remaining four models operate in continuous-time, each paralleling their discrete-time counterparts with a padding parameter $\kappa \in \{20, 40, 60, 80\}$. Further details on model architecture are provided in Appendix A.3. Models which include particle guidance described in Section 3.3 are denoted DiffER²PG and DiffER²PG+. In DiffER²PG, guidance sampling is run instead of the standard sampling procedure. In DiffER²PG+, guidance sampling is run alongside standard sampling procedures, resulting in an output distribution that combines DiffER² and DiffER²PG.

**Evaluation Metrics.** We evaluate our models using standard top-$k$ accuracy metrics. The DiffER² ensemble is run on each product in the test set and generated reactants are ranked by frequency, with the most generated reactant having the highest ranking. The top-$k$ accuracy is defined as the proportion of reactions in which the ground-truth reactant is in the top-$k$ ranks. Top-k accuracy is

---

[1]https://github.com/otori-bird/retrosynthesis

reported for $k \in \{1, 3, 5, 10\}$. We compare the DiffER$^2$ ensemble with a wide range of template-based, semi-template, and template-free methods for retrosynthesis prediction. However, we primarily focus on comparing performance against other template-free methods. In addition to top-k accuracy, we include results for round-trip accuracy using a forward synthesis model in Appendix **??** and provide a case study on a ring-forming reaction in the main results and three other reactions in Appendix A.8.

**Property-Specific Performance Analysis.** We additionally analyze how model performance compares against various structural properties of the SMILES sequences to better understand model performance. These properties include the target length difference between product and reactant SMILES, the edit distance between product and reactant SMILES, if the target reaction is ring adding/ring removing, and the number of branches in the target SMILES, each detailed in AppendixA.7. For each property, we calculate the slope and significance of a regression between the property statistic and the top-$k$ accuracy. In addition to these results, we report the performance of DiffER$^2$PG across different reaction types in Appendix A.6. We analyze how different molecular structures correlate with model performance in Appendix A.9. Finally, we analyze the attention maps for a non-ensemble diffusion model in Appendix A.10.

## 5 RESULTS

Top-K accuracies for the DiffER$^2$ ensemble are reported in Table 1. DiffER$^2$PG+ achieves the best top-1, top-3, and top-5 (statistically tied with R-SMILES) among template-free models and the second-best top-10 accuracy behind R-SMILES (Zhong et al., 2022). R-SMILES achieves the best top-5 and top-10 performance, but lags behind DiffER$^2$PG+ for $k = 1, 3$. DiffER$^2$PG+ may better estimate posterior peaks for SMILES sequences than R-SMILES, but it still struggles to sample from lower probability regions in the posterior. This result is paralleled in GDiffRetro (Sun et al., 2025), a semi-template method that utilizes diffusion sampling techniques. GDiffRetro achieves remarkable $k = 1$ accuracy, the highest of all baseline models, but struggles to produce strong results for higher values of $k$ compared to other methods. Sun et al. (2025) suggest this may be due to the oversampling of posterior peaks, which is also observed in our models. By employing an ensemble of diffusion models and incorporating SMILES-adapted particle guidance, DiffER$^2$PG+ can mitigate some of this bias and provide greater coverage of the posterior, as evidenced by our stronger performance for $k > 1$ compared to GDiffRetro. Compared to the original DiffER (Current et al., 2024), the DiffER$^2$ ensembles all achieve stronger results than the discrete-time only ensemble, particularly for higher values of $k$. This suggests that including continuous-time models greatly improves the diversity of output reactants. Notably, there is minimal difference between all DiffER methodologies for top-1 accuracy, indicating that the location of posterior peaks for the various ensembles may be similar, but that the shape of the distribution for lower probability samples is different. Interestingly, the top-k performances of DiffER$^2$ and DiffER$^2$PG are remarkably similar despite the latter ensemble producing more reactants.

Figure 2 shows box plots of the unique reactants generated by each ensemble for products in the test set. The greater number of unique molecules produced by DiffER$^2$PG indicates that the

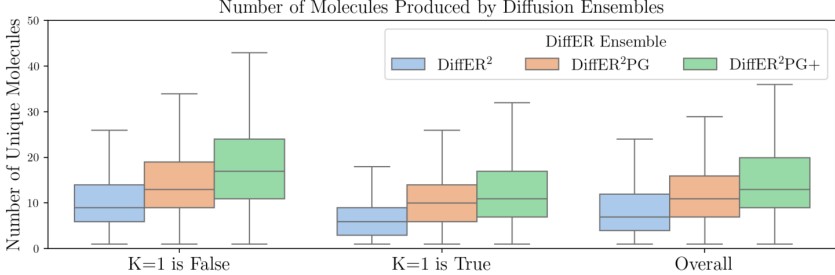

Figure 2: Box plot showing the number of unique molecules produced by the various DiffER$^2$ ensembles. "K=1 is False" indicates reactions that do not have an accurate top-1 reactant. "K=1 is True" indicates reactions that do have an accurate top-1 reactant. "Overall" includes all reactions.

Table 1: Top-K accuracy for a select set of baseline models. A full table of results can be found in Appendix A.4. The best performing model in each category is **bolded**, while the second-best performing model is underlined. A † indicates results that are statistically significant from DiffER at the 95% level.

| Category | Model | K=1 | 3 | 5 | 10 |
|---|---|---|---|---|---|
| Template-based | GLN (Dai et al., 2019) | 52.5 | 69.0 | 75.6 | 83.7 |
| | LocalRetro (Chen & Jung, 2021) | **53.4** | **77.5** | **85.9** | **92.4** |
| Semi-template | RetroPrime (Wang et al., 2021) | 51.4 | 70.8 | 74.0 | 76.1 |
| | $G^2$Retro (Chen et al., 2023) | 53.9 | 74.6 | 80.7 | **86.6** |
| | GDiffRetro (Sun et al., 2025) | **58.9** | **79.1** | **81.9** | - |
| Template-free | MEGAN (Sacha et al., 2021) | 48.1 | 70.7 | 78.4 | 86.1 |
| | R-SMILES (Zhong et al., 2022) | 56.3 | 79.2 | **86.2** | **91.0** |
| | DiffER (Current et al., 2024) | 57.6 | 79.0 | 84.1 | 87.4 |
| | $DiffER^2$ | 57.6 | 79.5 | 85.3† | 88.2 |
| | $DiffER^2$PG | 57.4 | 79.4 | 85.4† | 88.6† |
| | $DiffER^2$PG+ | **57.7** | **79.8** | 86.1† | 89.9† |

SMILES-adapted particle guidance methodology does result in greater output diversity and reduced sampling of posterior peaks. The combined result of $DiffER^2$PG+ has an even greater number of unique molecules, indicating that the difference between sample sets generated from $DiffER^2$ and $DiffER^2$PG is substantial. Notably, this result may be impacted by the limited number of samples taken from each model in the ensemble, which may not reflect the full posterior distribution for each model. Regardless, $DiffER^2$PG is an effective approach to improving output diversity while maintaining comparable performance to $DiffER^2$. This is desirable because multiple reactants could produce the same product, thus validating accuracy, and output diversity is critical. Like Current et al. (2024), we observe that fewer molecules are sampled when the top-1 result is accurate, indicating that the models still over-sample from posterior peaks to the deficit of output diversity. However, these effects are lessened compared to DiffER.

**Case Study.** A case study on an N-heterocyclization reaction with dihalides for the $DiffER^2$PG+ ensemble is presented in Figure 3. Additional case studies are presented in Appendix A.8. The target reaction is a reaction of a primary amine with a dihalide in an alkaline aqueous medium to form a heterocyclic ring (Ju & Varma, 2006). While the ensemble model predicts this reaction, it is only predicted 4% of the time, making it the third most frequently predicted reaction. The top-ranked predicted reaction is a Borch reductive amination (Borch, 2003) in which a carbonyl group is reacted with a secondary amine in the presence of a reducing agent to produce a tertiary amine. Unfortunately, due to the presence of multiple carbonyl groups in the predicted reactant, this reaction will likely produce unwanted byproducts. The second-ranked reaction is a substitution reaction with a mesylate ester and a secondary amide nucleophile. Again, this reaction does produce the desired product, but may produce byproducts due to the electrophilicity of the carbamate group, which may react with the amine nucleophile.

To recap, all three top-ranked reactions could be used to form the target product, but only the 3rd reaction, which matches the ground truth, would limit the creation of byproducts. Notably, this reactant is not produced by the original DiffER ensemble (Current et al., 2024), demonstrating that $DiffER^2$PG+ significantly improves output diversity to benefit performance. In this case study, as well as those presented in Appendix A.8, $DiffER^2$PG+ demonstrates significant capability to predict reactants that could be used to produce the desired product, though some reactions may be suboptimal due to the possibility of undesirable side products. Thus, while the models can learn appropriate reaction mechanisms and patterns, the predicted forward reaction's purity may not be considered by the models. Future work should consider approaches that include reducing the possibility of side products as an objective. Furthermore, ambiguity in the optimality of reactants could be mitigated by providing additional guidance to the model in a human-in-the-loop setting.

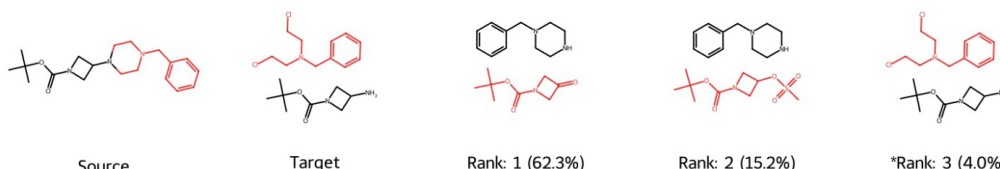

Figure 3: Case study on a N-heterocyclization reaction with dihalides. From left to right: the source product, the ground-truth reactant(s), and the top-3 predicted reactant(s) for the DiffER$^2$PG+ ensemble. Red highlighting indicates differences between the source product and the reactants. The value in parentheses indicates the rate at which the reactant(s) are produced by the ensemble.

**Property Results.** Figure 4 shows the results of the property-specific performance analysis described in Section 4 for top-1 accuracy, with additional results available in Appendix A.7. These results highlight substantial differences in the capabilities of DiffER$^2$PG+ and R-SMILES. DiffER$^2$PG+ is significantly more capable at predicting reactant SMILES with a larger increase in sequence length than the auto-regressive model. This result also helps explain the increased capability of diffusion models to predict ring-removing reactions, as these are often reactions with a high increase in sequence length. This difference in capability can be understood via the attention maps for the diffusion models: rather than building up the sequence token-by-token like an auto-regressive model, the diffusion models work backward, attending to the full-sequence length in early layers then decreasing the attended length as the model goes deeper (see Appendix A.10). Thus, diffusion models may be "less inclined" to produce shorter sequences.

In contrast, diffusion models seem less capable of modeling ring-adding reactions than auto-regressive models, though this difference is only significant for $k > 1$. Both models have a negative association with the edit distance between product and reactant SMILES, except for top-1 accuracy, in which DiffER$^2$PG+ has no significant relationship with the edit distance. This suggests that some reactions with large edits remain "easy" for DiffER$^2$PG+ to predict. Both models struggle to generate sequences with highly non-sequential behavior, as evidenced by the negative relationship between performance and branch count. Overall, DiffER$^2$PG+ outperforms R-SMILES conditioned on all properties except for ring-adding reactions.

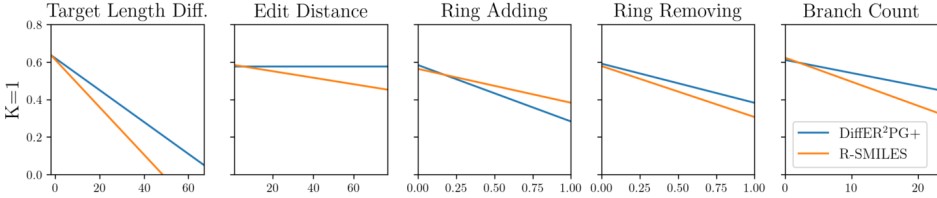

Figure 4: Lines of best fit between top-k accuracy and reactant properties for the DiffER$^2$PG+ and R-SMILES (Zhong et al., 2022) models. All slopes are significantly different from 0 ($p < 0.01$) except Edit Distance.

## 6 CONCLUSION

In this work, we introduce SMILES-adapted particle guidance to encourage output diversity in diffusion models for SMILES generation on the chemical retrosynthesis task. Combining this guidance mechanism with the ensemble approach of DiffER (Current et al., 2024), we can significantly increase the number of unique molecules generated while improving top-k accuracy metrics on benchmark datasets. Furthermore, we offer an in-depth empirical analysis on the capabilities and limitations of discrete diffusion for SMILES generation compared to traditional auto-regressive approaches. In future work, we aim to investigate additional guidance methods for SMILES diffusion models to reduce byproducts and explore applications of rectified linear flows for SMILES generation.

## REPRODUCIBILITY

We detail all model architectures and hyperparameters used to produce DiffER$^2$ in Appendix A.3. We follow the methods outlined in Section 3 and Appendix A.2 alongside the experimental setup in Section 4 to train and evaluate our models. All code used to produce the results of DiffER$^2$ will be made publicly available.

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
