## A APPENDIX

### A.1 DETAILS ON DIFFER'S PADDING PROCEDURE

During training, DiffER (Current et al., 2024) randomly appends padding tokens to the SMILES representations of reactants. The target reactant sequence $x_0^{1:L_x}$ is randomly padded on the right by a sequence of padding tokens with random length, such that

$$x_0' = \left[ x_0^{1:L_x}, e_p^{1:L_p} \right] \text{ and } x_t \sim q(x_t|x_0'), \tag{14}$$

where $e_p$ is the one-hot encoder vector of the pad token and $L_p \sim \mathcal{U}\{1, \kappa\}$, with $\mathcal{U}\{1, \kappa\}$ being the discrete uniform distribution between 1 and some constant $\kappa > 1 \in \mathbb{N}$. The choice of $\kappa$ allows the diffusion model to strike a balance between effective length prediction with $f_\ell^\phi$ and allowing $f_t^\theta$ to determine the sequence length by predicting padding tokens during sampling.

We can show that the variant padding methodology is in effect a moving average kernel on the distribution of $(L_x - L_y)$ with width $\kappa$. Let $L_{x-y} \in [-L/2, L/2]$ denote the difference $(L_x - L_y)$ with probability distribution $L_{x-y} \sim \mathcal{C}(L_{x-y}; p_\ell)$, where the probability vector $p_\ell$ is calculated from the training data distribution. Let $L'_{x-y} \sim \mathcal{C}(L'_{x-y}; p'_\ell)$ represent the difference distribution after random padding with probability vector $p'_\ell$, and let $P_{i|j} = P(L'_{x-y} = i|L_{x-y} = j)$ be the probability that a sequence pair with difference $j$ maps to a difference of $i$ for the padded sequence. Then

$$p'_{\ell i} = \sum_{j=1}^{L} P_{i|j} \cdot p_{\ell j} = \frac{1}{\kappa} \sum_{j=1}^{L} p_{\ell j} \mathbb{I}[i - \kappa \le j < i] = \frac{1}{\kappa} \sum_{j=i-\kappa}^{i-1} p_{\ell j}. \tag{15}$$

Thus, $p'_\ell$ is a moving average of the previous $\kappa$ indices of $p_\ell$. In effect, the padding methodology reduces the heavy bias observed in the distribution of $L_{x-y}$ by smoothing the probability vector $p_\ell$.

### A.2 TRAINING DETAILS FOR PARTICLE GUIDANCE MODEL

---

**Algorithm 1** Pseudocode for particle guidance sampling procedure.

---

**Inputs**: target product $y_0$, timesteps $T$, number of samples $N$, diffusion model encoder $f_{encoder}^\theta$, diffusion model decoder $f_{decoder}^\theta$, noise schedule $\bar{\alpha}_t$

**Output**: Sampled reactants $\{\hat{x}_0^{(1)}, \hat{x}_0^{(2)}, ..., \hat{x}_0^{(N)}\}$

1: **for** $i$ in 1 to $N$ **do**
2:     Initialize $y_0^{(i)} = \text{Augment}(y_0)$ according to Zhong et al.Zhong et al. (2022)
3:     Initialize $x_T^{(i)} = \mathcal{C}(x_T; m)$ from noise distribution
4:     Compute encoder memory $H_i = f_{encoder}^\theta(y_0^{(i)})$
5: **end for**
6: **for** $t$ in $T$ to 1 **do**
7:     $s = t - 1$
8:     Let $X_t = \{x_t^{(1)}, x_t^{(2)}, ..., x_t^{(n)}\}$
9:     **for** $i$ in 1 to $N$ **do**
10:         Predict $\hat{x}_0^{(i)} \leftarrow f_{decoder}^\theta(x_t^{(i)}, H_i, t)$
11:         Compute $p_\Phi(c = 1|x_t^{(i)}, X_t)$ according to equation 11
12:         Compute $x_s^{(i)}$ according to equations 12 and 13
13:         Update $f_{\mathcal{K}}^\Phi$ and $f_{encoder}^\theta$ using classifier loss (Binary Cross Entropy)
14:     **end for**
15: **end for**
16: $\hat{x}_0^{(i)} \leftarrow f_{decoder}^\theta(x_t^{(i)}, H_i, t)$
17: Return $\{\hat{x}_0^{(1)}, \hat{x}_0^{(2)}, ..., \hat{x}_0^{(N)}\}$

---

The guidance model $f_{\mathcal{K}}^\Phi$ is trained in a reinforcement learning paradigm during inference with no explicit pretraining. Recognizing that the encoder portion of the diffusion model $f_{encoder}^\theta$ is time-independent and already adept at recognizing molecular structure, we construct $f_{\mathcal{K}}^\Phi$ as an additional

classification head following an $f_{encoder}^\theta$. Let $H_i^{1:L_i} \in \mathbb{R}^{L_i \times d}$ be the hidden representation of a SMILES sequence $x^{(i)} \in \{0,1\}^{L_i \times K}$ with hidden size $d$ and length $L_i$ obtained from the encoder model, i.e., $H_i^{1:L_i} = f_{encoder}^\theta(x^{(i)})$. Let $H^{(i)} = \frac{1}{L_i} \sum_{\ell=1}^{L_i} H_i^{[\ell]}$ be the hidden representation average over the sequence length. The classifier model is constructed as a function of the difference between the mean hidden representations for strings $x^{(i)}, x^{(j)}$ along the sequence length:

$$f_{\mathcal{K}}^\Phi(x^{(i)}, x^{(j)}) = \sigma \left( \Phi^\top \left( H^{(i)} - H^{(j)} \right) \right), \tag{16}$$

where $\Phi \in \mathbb{R}^d$ are the model parameters. By averaging over the sequence length $L_i$ to construct $H^{(i)}$, we help remove the effect of sequence ordering on the classification. This is desirable due to non-singular nature of SMILES representations.

During inference, we train $f_{\mathcal{K}}^\Phi$ in a manner akin to reinforcement learning. Note that $f_{\mathcal{K}}^\Phi$ is not explicitly pretrained aside from the encoder model $f_{encoder}^\theta$. We allow both the classifier weights $\Phi$ and the encoder weights $\theta_{encoder}$ to be optimized during sampling. Note that because the encoder model is only run once at the beginning of sampling, changing the weights of $f_{encoder}^\theta$ during future sampling steps does not directly affect the overall diffusion model $f_t^\theta$. Pseudocode for the joint training/sampling procedure for $f_{\mathcal{K}}^\Phi$ is provided in algorithm 1.

## A.3 DETAILS ON MODEL ARCHITECTURES AND DIFFUSION PARAMETERS

All models in the DiffER$^2$ ensembles take the form of standard encoder-decoder transformers with 6 layers, 8 attention heads, a latent size of 512, a feed-forward layer of size 2048, and GeLU activation Hendrycks & Gimpel (2016). All models are trained using an Adam optimizer with learning rate 0.0001 and a dropout rate of 0.1 for 90 epochs. Both discrete-time and continuous-time models are trained using a log-linear noise schedule and are sampled over 100 diffusion timesteps. All models are trained on a single Nvidia A6000 GPU with 48GB memory.

## A.4 COMPLETE TABLE OF TOP-K ACCURACY

A full table of top-k metrics is provided in table 2. This table includes additional baseline approaches for computerized retrosynthesis prediction from prior literature as well as results for the singular C-DiffER and C-DiffER$_{20}$ diffusion models. These results highlight the advantages of the ensemble approaches, which produce sample distributions with higher top-k accuracy than the singular diffusion models. Notably, C-DiffER does achieve stronger performance than C-DiffER$_{20}$ (which includes the DiffER adapted length prediction described in section 3.2), demonstrating the inherent bias in the length prediction model, but still shows reduced performance compared to R-SMILES, indicating that a single diffusion model may be an insufficient approximation of the reactant posterior. These results may also change if a greater number of samples were taken from the C-DiffER models.

C-DiffER$_{Selfies}$ indicates the C-DiffER model trained using SELFIES molecular representations (Krenn et al., 2019), an alternative to SMILES string representations. Unlike SMILES, SELFIES is 100% robust, with every SELFIES string representing a valid molecular structure. SELFIES has been proposed as an alternative to SMILES for molecular generation tasks due to the guaranteed validity of outputs, which do not suffer from the harsh syntactic constraints of SMILES. Notably, however, the SELFIES-based implementation of C-DiffER does not seem to significantly improve performance compared to SMILES-based models, and in fact exhibits a lower top-1 accuracy than the SMILES-based version. As $k$ increases, the performances become more similar, with the SELFIES-based implementation exhibiting a slightly stronger top-10 accuracy. This indicates that the choice of string-encoding has little impact on the performance of the retrosynthetic diffusion model. This is likely due to the already high per-sample validity (89.4%) of SMILES strings produced by C-DiffER, which limits the impact of SELFIES' guaranteed validity on the model's performance. Furthermore, these results support the findings of Leon et al. (2024), which found that SMILES-based models often perform better than their SELFIES counterparts on a variety of molecular prediction tasks.

Table 2: Complete top-K accuracy comparison to baseline models. DiffER$^2$ is the ensemble model and C-DiffER are the individual continuous-time diffusion models. The best performing model in each category is **bolded**, while the second-best performing model is underlined.

| Category | Model | K=1 | 3 | 5 | 10 |
|---|---|---|---|---|---|
| Template-based | Retrosim Coley et al. (2017) | 37.3 | 54.7 | 63.3 | 74.1 |
| | Neuralsym Segler & Waller (2017) | 44.4 | 65.3 | 72.4 | 78.9 |
| | GLN Dai et al. (2019) | 52.5 | 69.0 | 75.6 | 83.7 |
| | LocalRetro Chen & Jung (2021) | **53.4** | **77.5** | **85.9** | **92.4** |
| Semi-template | G2Gs Shi et al. (2020) | 48.9 | 67.6 | 72.5 | 75.5 |
| | GraphRetro Somnath et al. (2021) | 53.7 | 68.3 | 72.2 | 75.5 |
| | RetroXpert Yan et al. (2020) | 50.4 | 61.1 | 62.3 | 63.4 |
| | RetroPrime Wang et al. (2021) | 51.4 | 70.8 | 74.0 | 76.1 |
| | G$^2$Retro Chen et al. (2023) | 53.9 | 74.6 | 80.7 | **86.6** |
| | GDiffRetro Sun et al. (2025) | **58.9** | **79.1** | **81.9** | - |
| Template-free | Seq2Seq Liu et al. (2017) | 37.4 | 52.4 | 57.0 | 61.7 |
| | Levenshtein Sumner et al. (2020) | 41.5 | 48.1 | 50.0 | 51.4 |
| | GTA Seo et al. (2021) | 51.1 | 67.6 | 74.8 | 81.6 |
| | Graph2SMILES Tu & Coley (2022) | 51.2 | 66.3 | 70.4 | 73.9 |
| | Dual-TF Sun et al. (2021) | 53.3 | 69.7 | 73.0 | 75.0 |
| | MEGAN Sacha et al. (2021) | 48.1 | 70.7 | 78.4 | 86.1 |
| | Chemformer Irwin et al. (2022) | 54.3 | - | 62.3 | 63.0 |
| | Retroformer Wan et al. (2022) | 53.2 | 71.1 | 76.6 | 82.1 |
| | Tied transformer Kim et al. (2021) | 47.1 | 67.2 | 73.5 | 78.5 |
| | R-SMILES Zhong et al. (2022) | 56.3 | 79.2 | **86.2** | **91.0** |
| | DiffER Current et al. (2024) | 57.6 | 79.0 | 84.1 | 87.4 |
| | C-DiffER | 55.9 | 73.3 | 75.8 | 76.8 |
| | C-DiffER$_{20}$ | 54.5 | 72.6 | 75.6 | 76.5 |
| | C-DiffER$_{Selfies}$ | 54.3 | 72.3 | 75.2 | 77.5 |
| | DiffER$^2$ | 57.6 | 79.5 | 85.3 | 88.2 |
| | DiffER$^2$PG | 57.4 | 79.4 | 85.4 | 88.6 |
| | DiffER$^2$PG+ | **57.7** | **79.8** | 86.1 | 89.9 |

## A.5 ROUND-TRIP ACCURACY

to further evaluate our models, we compute the top-k round-trip accuracy score using the forward model of R-Smiles (Zhong et al., 2022). The round-trip accuracy is calculated by running a forward synthesis prediction model on the predicted reactants of the retrosynthesis model and computing the rate at which the original input product is recovered. We use the forward prediction model of R-Smiles, which achieves a top-1 accuracy of 91% on the USPTO-50K dataset. However, we note that the accuracy of the forward model is likely lower for retrosynthesis predictions due to the lack of reagent predictions in the retrosynthesis benchmark, which are generally included as input to the forward model. Due to the lack of reagent information amongst retrosynthesis models, the forward model must make assumptions about reagents used, possibly reducing the performance of the model when applied to the outputs of present retrosynthesis models.

Table 3: Top-K round-trip accuracy for select models. A † indicates results that are significantly different from DiffER at the 95% level.

| Model | K=1 | 3 | 5 | 10 |
|---|---|---|---|---|
| R-SMILES Zhong et al. (2022) | 53.8 | 66.7 | 73.7 | 82.1 |
| DiffER Current et al. (2024) | 53.8 | 64.7 | 70.1 | 75.6 |
| DiffER$^2$ | 54.1 | 65.2 | 70.6 | 75.5 |
| DiffER$^2$PG | 54.4 | 66.0 | 71.8 | 77.6 |
| DiffER$^2$PG+ | 54.4 | 66.1 | 72.0$^\dagger$ | 78.4$^\dagger$ |

The top-k round-trip accuracy results are presented in table 3. These scores support the conclusions obtained from the ground-truth top-k accuracy presented in table 1: DiffER$^2$PG+ consistently obtains the highest accuracy scores among the proposed models, significantly improving upon DiffER for $k = 5$ and $k = 10$. Furthermore, DiffER$^2$PG+ maintains the best top-1 accuracy among the tested models, though the round-trip accuracies for $k = 1$ differ only slightly. Notably, the difference in performance between DiffER$^2$ and DiffER$^2$PG is significantly greater when analyzing round-trip accuracy versus ground-truth accuracy, indicating that the output distribution obtained using SMILES-adapted particle guidance sampling maintains not only improves output diversity but also predicts quality reactions. These results further support the conclusion that the proposed DiffER$^2$ changes significantly improve the output diversity and efficacy of diffusion models for template-free retrosynthesis prediction.

## A.6 REACTION TYPE PERFORMANCE

Table 4: Top-K accuracy metrics by reaction type for the DiffER$^2$PG+ ensemble. Support indicates the number of reactions of the associated type in the test set.

| Reaction Type | K=1 | 3 | 5 | 10 | Support |
|---|---|---|---|---|---|
| Heteroatom alkylation and arylation | 58.5 | 81.1 | 87.5 | 91.2 | 1516 |
| Acyclation and related processes | 70.1 | 89.9 | 93.5 | 95.5 | 1190 |
| Deprotections | 55.2 | 80.8 | 87.1 | 90.5 | 822 |
| C-C bond formation | 41.3 | 64.9 | 73.9 | 81.5 | 567 |
| Reductions | 56.6 | 78.4 | 85.5 | 89.4 | 463 |
| Functional group interconversion | 39.6 | 56.0 | 70.3 | 75.3 | 182 |
| Heterocycle formation | 32.2 | 52.2 | 63.3 | 66.7 | 90 |
| Oxidations | 60.2 | 81.9 | 86.7 | 91.6 | 83 |
| Protections | 70.1 | 89.6 | 94.0 | 95.5 | 67 |
| Functional group addition | 78.3 | 87.0 | 87.0 | 91.3 | 23 |

Table 4 reports the results of the DiffER$^2$PG+ ensemble across reaction types as determined by Schneider et al. Schneider et al. (2016). Notably, the performance of the ensemble model is inconsistent across reaction types. This has been similarly observed in other literature Chen et al. (2023); Current et al. (2024), and is an unsurprising result due to the differing degree of structural changes across different reaction types. The performance of the ensemble model for is lowest for heterocycle formation, functional group interconversion, and C-C bond formation. While all three reaction types exhibit low top-1 accuracy ($< 40\%$), functional group interconversion and C-C bond formation are able to achieve moderately strong results ($> 75\%$) for top-10 accuracy, a significant increase in performance. This suggests that the models are capable of learning reaction patterns associated with these reaction types, but may be unable to discern when they are appropriate to be used without additional guidance. In contrast, heterocycle formation still achieves limited performance ($\sim 65\%$ accuracy) for $k = 10$, indicating that the reaction patterns of heterocycle formation may be less understood by the diffusion models.

In all three cases, much of the difficulty in predicting patented reactions may be attributed to ambiguity in the goals of the single-step retrosynthesis task. Unlike multi-step retrosynthetic planning, which aims to produce a reaction pathway from a core set of available reactions, single-step retrosynthesis models are generally unaware of the availability of reactants. Furthermore, as exemplified in the case studies later presented in appendix A.8, there are multiple possible reactants that could be used to produce the target product, and ambiguity in why one reactant may be preferential over another makes training a capable retrosynthesis model a daunting task. If we can develop methods to better inform the retrosynthesis model when one reaction may be beneficial over another, we may be better able to construct robust and useful retrosynthesis predictors.

In contrast to the poor performance of heterocycle formation, functional group interconversion, and C-C bond formation, all other reactions types achieve strong ($\sim 90\%$ or greater) top-10 accuracy, with some even achieving considerable top-1 accuracy ($> 70\%$). Many of these reaction types utilize common or routine reactions: functional group addition, for instance, has extremely low diversity in reactants. While the target product is different for each sample, the vast majority of functional

group additions in the test set are bromination reactions featuring N-bromosuccinimide as a reactant, a commonly used reagent for its bromine-donating properties. This commonality of reactants is also observed in the protections subset. In contrast, the high-support acyclation and related processes and heteroatom alkylation and arylation subsets exhibits a wider diversity of reactants, but all with a shared mechanism which is effectively learned by the ensemble models, with the main point of decision being where to "split" the product. As there are generally few locations where small molecules may be "split" effectively, these reaction subsets will tend toward higher performance, particularly as $k$ increases. Overall, reaction types that are learned well by the DiffER$^2$PG+ tend to have consistent and common reaction patterns with limited possible locations for a reaction center.

## A.7 Property Coefficient Analysis for DiffER$^2$ Models

In this appendix, we offer an analysis of the DiffER$^2$ models by regressing model performance with a variety of target reaction properties. These properties were selected due to the insight they give into the comparable and contrasting capabilities of diffusion and auto-regressive models on SMILES reactions. These properties are:

1. **Target Length Difference**: the difference in sequence length $L_x - L_y$ between the product SMILES and the target reactant SMILES. This property was chosen to characterize differing capabilities between diffusion models and auto-regressive models and can serve as a proxy for the leaving group size.

2. **Edit Distance**: the Levenshtein distance between the product and reactant SMILES. This property characterizes how sequence similarity between the product and reactant affects performance.

3. **Ring Adding**: If the number of rings in the product is greater than the number of rings in the reactant. This is indicative of ring formation. This represents a significant change in molecular and SMILES structure.

4. **Ring Removing**: If the number of rings in the product is less than the number of rings in the reactant. Oftentimes, this is indicative of a large leaving group containing a ring structure rather than a true ring opening. This represents a significant change in molecular and SMILES structure.

5. **Branch Count**: the number of branches in the target reactant SMILES. This is computed as the number of '(' tokens in the SMILES string. This represents structural complication in the SMILES string and molecular structure.

For each property, we report the slope of a line of best fit between the property statistic and the top-$k$ accuracy. A positive slope indicates that the model tends to perform well when the property statistic is high, and a negative slope indicates worse performance. In addition, we report the significance of each coefficient compared to the coefficients achieved by Root-aligned SMILESZhong et al. (2022) to highlight significant differences between auto-regressive and diffusion models. This is accomplished by constructing the z-statistic $z = (\beta_{DiffER} - \beta_{RSmiles})/\sqrt{SE_{DiffER}^2 + SE_{RSmiles}^2}$, where $\beta$ is the slope and $SE$ is the standard error Clogg et al. (1995). These results are reported for the DiffER$^2$ ensemble and the continuous-time C-DiffER and C-DiffER$_{20}$ models in table 5, with a visualization of the best fit lines for each property provided in figure 5.

Table 5 reports the slope coefficients, significance, and difference in coefficient values compared to R-SMILES for each property for each of the DiffER$^2$ models as well as the C-DiffER models. Figure 5 offers a visualization of the lines of best fit for $k = 1$, $k = 3$, and $k = 10$ performance conditioned on each of the analyzed properties. The results for $k = 3$ and $k = 10$ support the same conclusions discussed in the main text for $k = 1$, primarily that:

1. Diffusion models with length prediction are significantly more capable at predicting ground-truth reactions with larger increases in reactant SMILES length from the product SMILES compared to auto-regressive methods;

2. Diffusion ensembles are significantly more capable at predicting ring-removing reactions compared to auto-regressive methods, but struggle to predict ring-adding reactions;

Table 5: Coefficients between reaction properties and performance metrics for the $DiffER^2$ ensemble and the C-DiffER model. The value in parenthesis indicates the change in coefficient compared to coefficients obtained using R-Smiles Zhong et al. (2022). Values that are significantly different from 0 are underlined at the 95% level and **bolded** at the 99% level.

| Property | Model | K=1 | 3 | 10 |
|---|---|---|---|---|
| Target Length Diff. | C-DiffER | **-0.0092** (+0.0035) | **-0.0108** (+0.0023) | **-0.0104** (+0.0024) |
| | $C\text{-}DiffER_{20}$ | **-0.0071 (+0.0056)** | **-0.0084 (+0.0047)** | **-0.0072 (+0.0055)** |
| | $DiffER^2$ | **-0.0078 (+0.0049)** | **-0.0090 (+0.0041)** | **-0.0066 (+0.0061)** |
| | $DiffER^2PG$ | **-0.0074 (+0.0052)** | **-0.0075 (+0.0056)** | **-0.0060 (+0.0067)** |
| | $DiffER^2PG+$ | **-0.0085** (+0.0041) | **-0.0090 (+0.0041)** | **-0.0057 (+0.0070)** |
| Edit Distance | C-DiffER | 0.0003 (+0.0020) | **-0.0020** (+0.0007) | **-0.0024** (+0.0000) |
| | $C\text{-}DiffER_{20}$ | 0.0003 (+0.0021) | **-0.0021** (+0.0007) | **-0.0016** (+0.0008) |
| | $DiffER^2$ | 0.0000 (+0.0017) | **-0.0019** (+0.0009) | **-0.0015** (+0.0010) |
| | $DiffER^2PG$ | 0.0005 (+0.0022) | -0.0014 (+0.0014) | **-0.0016** (+0.0009) |
| | $DiffER^2PG+$ | 0.0000 (+0.0017) | **-0.0019** (+0.0009) | **-0.0015** (+0.0010) |
| Ring Adding | C-DiffER | **-0.2823** (-0.1016) | **-0.2895** (-0.0601) | **-0.3083** (-0.1189) |
| | $C\text{-}DiffER_{20}$ | **-0.2592** (-0.0786) | **-0.3513** (-0.1219) | **-0.3400 (-0.1506)** |
| | $DiffER^2$ | **-0.2826** (-0.1020) | **-0.3274** (-0.0980) | **-0.3571 (-0.1677)** |
| | $DiffER^2PG$ | **-0.2897** (-0.1091) | **-0.3867 (-0.1574)** | **-0.3273 (-0.1379)** |
| | $DiffER^2PG+$ | **-0.3009** (-0.1203) | **-0.3565** (-0.1271) | **-0.3320 (-0.1426)** |
| Ring Removing | C-DiffER | **-0.2316** (+0.0403) | **-0.2397** (+0.0048) | **-0.2258** (-0.0136) |
| | $C\text{-}DiffER_{20}$ | **-0.1891** (+0.0828) | **-0.1782** (+0.0662) | **-0.1656** (+0.0466) |
| | $DiffER^2$ | **-0.2166** (+0.0553) | **-0.1821** (+0.0624) | **-0.1397 (+0.0726)** |
| | $DiffER^2PG$ | **-0.1878** (+0.0841) | **-0.1453 (+0.0992)** | **-0.0957 (+0.1166)** |
| | $DiffER^2PG+$ | **-0.2088** (+0.0631) | **-0.1585 (+0.0860)** | **-0.0854 (+0.1268)** |
| Branch Count | C-DiffER | **-0.0079** (+0.0050) | **-0.0106** (+0.0023) | **-0.0132** (-0.0020) |
| | $C\text{-}DiffER_{20}$ | **-0.0081** (+0.0048) | **-0.0106** (+0.0024) | **-0.0098** (+0.0015) |
| | $DiffER^2$ | **-0.0071** (+0.0058) | **-0.0100** (+0.0030) | **-0.0070** (+0.0043) |
| | $DiffER^2PG$ | **-0.0073** (+0.0055) | **-0.0085** (+0.0044) | **-0.0082** (+0.0031) |
| | $DiffER^2PG+$ | **-0.0069** (+0.0059) | **-0.0094** (+0.0036) | **-0.0070** (+0.0043) |

3. Both diffusion models and auto-regressive models are negatively impacted by an increased edit distance between product and reactant SMILES as well as an increased branch count in the target reactant.

Notably, however, many of the observed performance differences between R-SMILES and the $DiffER^2$ models become more significant at larger $k$, indicating that the trends observed at $k = 1$ remain consistent across when more unique molecules are generated by the models. Furthermore, this highlights the capability for both model archetypes to predict common reaction rules that are likely to be used with high probability, resulting in $k = 1$ distributions that perform more similarly than the more diverse $k = 10$ molecule sets.

One interesting difference between $DiffER^2PG+$ and R-SMILES that becomes more significant at $k = 10$ is the effect of branch count on model performance. While $DiffER^2PG+$ has higher slope for all values of $k$, the difference is only significant for $k = 10$, as shown in Table 5. This difference in performance however is better observed in Figure 5, which also shows the difference in bias between $DiffER^2PG+$ and R-SMILES. Notably, when the branch count is low, R-SMILES performs slightly better than $DiffER^2PG+$, indicating the auto-regressive model may be stronger when the reactant SMILES has a highly sequential nature. However, as the number of branch terms in the reactant SMILES increases, the non-auto-regressive diffusion ensembles begin to exhibit stronger performance. This suggests that the diffusion models may be slightly more capable of generating SMILES sequences with non-sequential properties compared to state-of-the-art auto-regressive models.

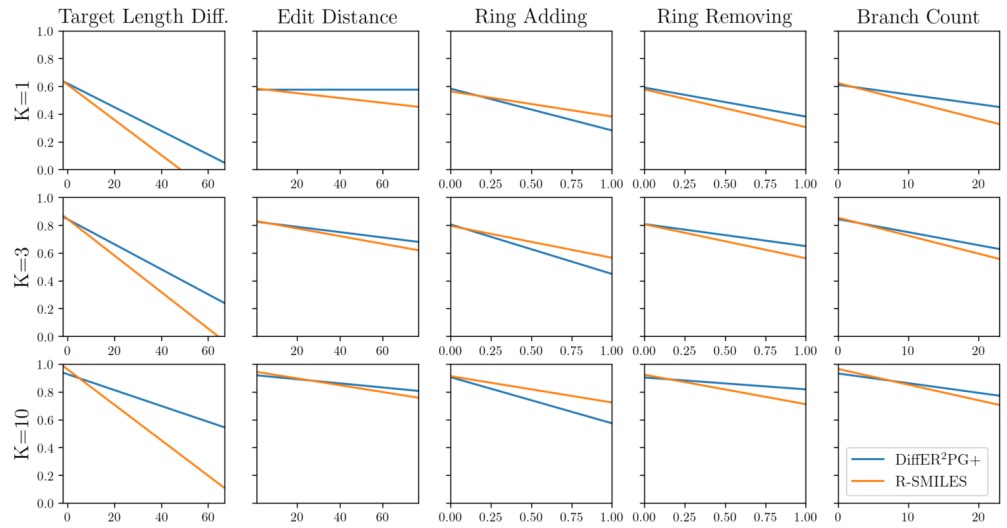

Figure 5: Lines of best fit between model performance and reactant properties for the DiffER$^2$PG+ and R-SMILES Zhong et al. (2022) models.

Table 6: Case Studies. From left to right: the source product, the ground-truth reactant(s), and the top-3 predicted reactant(s) for the DiffER$^2$PG+ ensemble. Red highlighting in the source molecule indicates the ground truth change. Red highlighting in the true and predicted reactant(s) indicates differences between the source product and the reactants.

Case 1: Nitro Reduction to Aniline

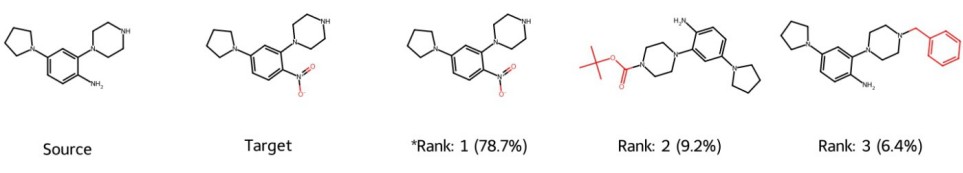

Case 2: Carbonyl Reduction to Alcohol

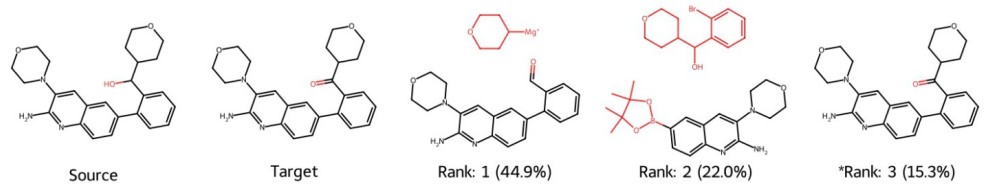

Case 3: Deprotection using Methoxymethyl Ether

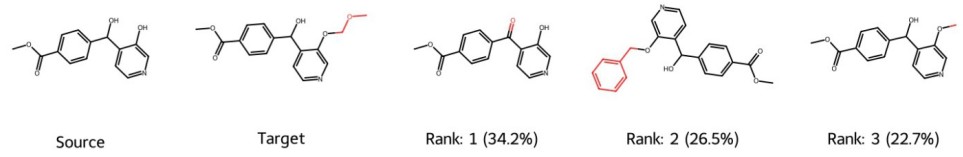

A.8    CASE STUDY

In this section, we present three additional case studies on the DiffER$^2$PG+ ensemble. These reactions were selected to demonstrate the strengths and weakness of the ensemble. Table 6 displays the source product, target reactant(s), and top-3 reactant(s) predicted by the DiffER$^2$PG+ ensemble.

**Case 1: Nitro Reduction to Aniline.**    Case 1 presents a reduction of a nitro group (O=N-O$^-$) to a primary aromatic amine (NH$_2$). This is a common reaction that is generally achieved through metal catalyzed reduction of the nitro group, often using Pd, Zn, Fe, or Ni. This same reaction is the most commonly predicted reaction by the ensemble model, encompassing 78.7% of all generated reactions. Instead of the nitro reduction, the second and third ranked reactions both take the form of an N-dealkylation to form a secondary amine. The rank 2 reaction is a proposed desterification in which the tert-butyl ester is reacted with a strong base such as NaOH to form an intermediate carboxylic acid and subsequently heated to decarboxylize, resulting in the target product. This is a viable alternative to the target reaction. The rank 3 reaction may be accomplished by Pt-catalyzed reaction with O$_2$ or by acid hydrolysis with hydrobromic or hydroiodic acid Najmi et al. (2022). This reaction is also a viable alternative to the target reaction. Each of the top-3 reactions proposed by the ensemble is a viable way to synthesize the desired product. Reactions may be limited by the availability of the generated reactant(s).

**Case 2: Carbonyl Reduction to Alcohol.**    Case 2 presents a common carbonyl reduction to an alcohol, in which a carbonyl group (C=O) is reacted with LiAlH$_4$ or NaBH$_4$ to form an alcohol (C-OH). This reaction is the third-most popular reaction produced by the ensemble model, appearing 15.3% of the time. In contrast to the target reaction, The proposed rank 1 and rank 2 reactions add larger structural changes to core molecule, and may be reactions that could have been used during prior synthesis of the ground truth reactant. The rank 1 reaction is an aldehyde reaction with a Grignard reagent. This is a common and viable method to produce the product molecule. The rank 2 reaction is a proposed Suzuki cross-coupling, in which a boronic ester (R$_1$-B(OR)$_2$) is reacted with an organohalide (R$_2$-X) using a Palladium catalyst in the presence of a base to produce the target product (R$_1$-R$_2$). The Suzuki coupling may also be a viable alternative to the ground truth reaction. Like case 1, each of the top-3 reactants may be suitable methods to synthesize the desired product, limited primarily by the availability of reactants.

**Case 3: Deprotection using Methoxymethyl Ether.**    Finally, case 3 presents a deprotection reaction in which a Methoxymethyl ether is removed via a silica-supported sodium hydrogen sulfate catalyst in dichloromethane solution Ramesh et al. (2003). Unfortunately, this reaction is not one of the top-3 reactions predicted by the ensemble, instead being produced at rank 7. The rank 1 prediction of the ensemble is a hydrolysis reaction in which a ketone is reduced to an alcohol, and is generally considered a common and trivial reaction. The rank 2 reaction is also a deprotection reaction involving a benzyl ether, which is removed using a metal catalyst in a hydrogen atmosphere. This is also a common and trivial reaction. Finally, the rank 3 predicted reaction is a acidic cleavage of a methyl ester. However, because this reaction is often done in an acidic environment, the presence of a ester group in the product/reactant may lead to unwanted byproducts. This reaction is non-trivial and is likely suboptimal compared to the rank 1, rank 2, and ground-truth reactions.

These case studies demonstrate both the effectiveness and pitfalls of the DiffER$^2$PG+ ensemble. As evidenced in all cases, the diffusion ensemble is capable of learning many common reaction patterns, but struggles to predict the ground-truth when many possible alternatives are present. While the reactions proposed by the DiffER$^2$ ensembles will generally produce the target, the model does not necessarily account for the availability of reactants or for the possible formation of side products, which may limit the models effectiveness. Furthermore, as evidenced by cases 2 and 3, it is not always clear what type of reaction is being desired: the model may be prone to larger structural changes (such as the top-2 reactants of case 2) over smaller changes that are necessary for preparing the molecule for future synthesis steps, such as protections, deprotections, and reductions. Moreover, the model may be biased toward reaction patterns that result in "easier" structural changes of the molecule, such as the top-2 reactions in the main case study (Figure 3), in which the ensemble prefers reactants in which two structures are "added" together over the more "advanced" ring-forming reaction. Nevertheless, for each product in the case studies, DiffER$^2$PG+ does generate the target reactant(s) within the top-10 predictions, and in many cases provides viable alternate reactions.

These case studies demonstrate that the diffusion ensemble is capable of learning effective reaction processes.

## A.9 PERFORMANCE ANALYSIS WITH MOLECULAR FINGERPRINTS

Morgan fingerprints Morgan (1965) are a commonly used methodology for constructing feature vectors from molecule structures. The core of the method is built around substructure matching: each feature vector is a set number of bits, which each bit in the feature indicating whether or not a specific substructure is present in the molecule. These substructures capture the environment of atoms in a set radius and are generated using a hashing algorithm provided by the original authors Morgan (1965). Because Morgan fingerprints capture many important characteristics of a molecule structure, we aim to investigate how the performance of the DiffER$^2$ models may be impacted by the presence of specific fingerprints in the target reactant(s), which could highlight specific substructures the ensembles may be biased against. Additionally, we note that because Morgan fingerprints are constructed using a hashing function, Morgan bits are not necessarily unique indicators, as multiple substructures may map to a single index. However, they are still a useful tool for visualizing how performance relates to particular substructures.

We generate Morgan fingerprints with 2048 bits and a radius of 2 for the target reactants and analyze the correlation between the top-10 performance of the DiffER$^2$PG+ model and each bit in the generated fingerprint. We visualize the Morgan bits that have the most positive correlation with model performance as well as the Morgan bits that have the most negative correlation with model performance. All analyzed correlations are statistically significant at the 99% ($p < 0.01$) level. Visuals are produced by randomly selecting a reactant from the test set that has the associated fingerprint bit and visualizing the atom and neighborhood that produced the positive bit value. Atoms highlighted blue represent the central atom, while atoms highlighted yellow represent atoms in an aromatic ring. Atoms with gray highlights are in aliphatic rings.

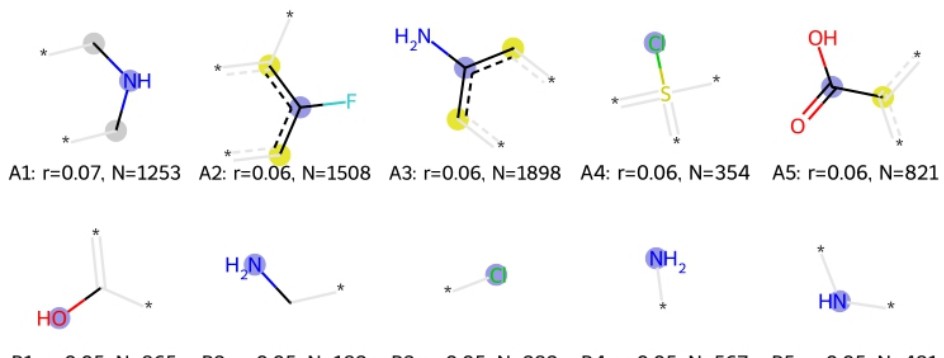

Figure 6: The 10 Morgan fingerprints with the highest positive correlation to Top-10 accuracy for the DiffER$^2$PG+ ensemble. "r" is the correlation value and "N" is the number of reactants in the test set that have the corresponding fingerprint. All correlations are statistically significant ($p < 0.01$).

Figure 6 visualizes the Morgan fingerprints that have the most positive correlation with model performance for $k = 10$. These bits tend to be substructures with well-known and highly utilized reaction mechanisms and reactivity rules. For instance, fingerprint **A1** has the highest correlation with model performance, and represents an aliphatic secondary amine. Reactions with aliphatic secondary amines are well known and highly studied in organic synthesis, appearing in $\sim 25\%$ of the test set. These reactions are commonly alyklation reactions, in which the amine is reacted with an alkyl halide to form new bonds and larger structures. Many fingerprints containing amines also exhibit positive correlation with model performance (**A3**, **B2**, **B4**, **B5**). Other notable fingerprints with positive correlation include: aryl halides (**A2**), which are commonly used in substitution reactions and as directing groups; carboxylic acids (**A5**) and alcohols (**B3**), two of the most studied functional groups in organic chemistry; halides (**B3**); and sulfonyl chlorides (**A4**), all of which are commonly used groups in organic synthesis.

In contrast, fingerprints that have significant negative correlation with model performance tend to be rarer, have ambiguity in the reaction, or introduce complexity in the SMILES sequence structure. Figure 7 visualizes the Morgan fingerprints that have the most negative correlation with model performance for $k = 10$. Many of these structures have a large number of branches or structural complexity (**C3**, **C5**, **D2**, **D3**, **D4**, **D5**), requiring SMILES structures with high non-sequential behavior. Many of these fingerprints may also require SMILES strings which detail stereochemistry of the molecule (**D1**, **D2**), further compounding the complexity of the SMILES string. In other cases, the fingerprint may require complexity in the SMILES string, but rather may represent ambiguity in the reactant. This is likely the case with **C2**, which represents a halide in the form of iodine. For many reactions, the explicit halide used may be interchangeable between Cl, Br, and I, as many chemical properties are shared between the three halogens with only subtle differences in reactivity and stability. A similar ambiguity may explain **C4**. Many reactions with alkenes result in the elimination of the C=C bond and the addition of new functional groups. To "reverse" this reaction, the retrosynthesis model would be have to identify a specific C-C bond as the reaction center. Because C-C bonds are extremely common, the model may have difficulty choosing these bonds as reaction centers, opting instead to target groups that are more commonly used in reactions. This behavior is readily observed in Appendix A.6, Table 4, which shows that the DiffER$^2$PG+ ensemble exhibits a reduced capability to predict reactions which form C-C bonds.

### A.10 ATTENTION MAP ANALYSIS

Our final results are an analysis of the attention maps for a single continuous-time diffusion model with full sequence padding (C-DiffER). This model was used to better understand how the attention of the model changes throughout the diffusion process without the bias of a length-prediction component as described in Section 3.2. We provide visualizations of the attention maps for both the self attention and attention on the input product for each layer in the transformer model for $t = 0.5$. Additionally, we map the attention on specific atoms in the SMILES sequences to the atoms they represent in the input and output molecular structures.

Figures 8 and 9 show the attention maps for layers 1, 2, 3 and 4, 5, 6, respectively for a amine alkylation reaction. This reaction reacts an alkyl halide with an amine and is successfully produced as a top-1 reactant by the C-DiffER model. These attention maps demonstrate how the construction of the SMILES sequence differs from auto-regressive methods. Rather than building the sequence from the beginning, the attention maps suggest that the diffusion model works backward, attending to the full-length sequence initially and decreasing the attended sequence length as the model goes deeper. This helps explain why the diffusion models may have a stronger affinity for generating reactants with a larger change in sequence length over the product SMILES: unlike auto-regressive models, which may be likely to predict a STOP token earlier in the sequence, diffusion models attend to the maximum possible sequence at each step in the diffusion process, allowing for the full sequence to be predicted over during each estimate of $x_0$. As the iteration number increases, the diffusion model

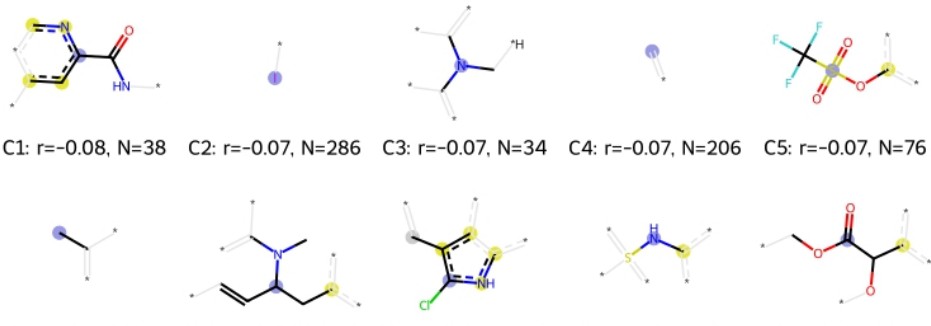

C1: r=-0.08, N=38    C2: r=-0.07, N=286    C3: r=-0.07, N=34    C4: r=-0.07, N=206    C5: r=-0.07, N=76

D1: r=-0.06, N=371    D2: r=-0.06, N=48    D3: r=-0.06, N=43    D4: r=-0.06, N=93    D5: r=-0.06, N=113

Figure 7: The 10 Morgan fingerprints with the highest negative correlation to Top-10 accuracy for the DiffER$^2$PG+ ensemble. "r" is the correlation value and "N" is the number of reactants in the test set that have the corresponding fingerprint. All correlations are statistically significant ($p < 0.01$).

becomes more confident in its output of padding tokens at later timesteps, attending to those positions less after the first couple layers.

In the early self attention layers, the diffusion model appears to try and understand the general structure of the SMILES string, estimating the level of noise and confidence in the input $x_t$. This can be observed in the way sequence attention is spread across the entire sequence for each token rather than focused on tokens with similar locality. After the first layer, the self attention mechanism places much more emphasis on nearby positions, but still attends very little to each tokens' own position, indicating that the model is not confident in the fidelity of a one-to-one input mapping between $x_t$ and $x_s$. Interestingly, in the final self attention layer, significant attention is placed around the reaction centers in the generated reactants. Source attention layers tend to place attention near non-carbon atoms in the target product early on, expanding to carbon atoms only in the later layers. While this may simply be an indication that these atoms are "unique" in the molecule compared to the carbon atoms, they may also be used as "anchors" to build the generated reactant(s) around. This could also explain why the models struggle to predict C-C bond formation reactions, as it would require the model to identify a common C-C bond as a reaction center, which may not have any significant uniqueness compared to other structures in the product molecule.

Another interesting feature of note is the repetitive attention patterns the predicted padding tokens place on the end of the product SMILES, notable in layers 2 and 3. Rather than the padding tokens placing attention on a single end token, the attention is stratified in a diagonal formation in groups of 10-12 tokens, attending primarily to the last 10-12 tokens of the product SMILES. This pattern is also observed in attention maps for other reactions, suggesting consistency in this behavior, which fades in the last couple layers. Repetition is a common phenomena also observed in large language models, which may repeat during inference time. The model only begins to reduce attention on later timesteps after this repetition begins, suggesting that it may use the repetition to "detect" padding locations. In the final layers of the transformer, attention on the source sequence is rather sparse and closely matches the attention maps observed by R-SMILES Zhong et al. (2022), with the attention patterns "matching" tokens between the input and output SMILES.

The attention map analysis offers some insight into the workings of the diffusion models, and may help inform future model creation. While we offer some theories behind the meaning of these attention maps as it relations to SMILES generation for chemical retrosynthesis, significantly more investigation and experimentation needs to be done before our theories can be confirmed. We hope these attention maps provide some insight into the capabilities and methods of transformer diffusion models and promote further research into the explainability of transformer diffusion models.

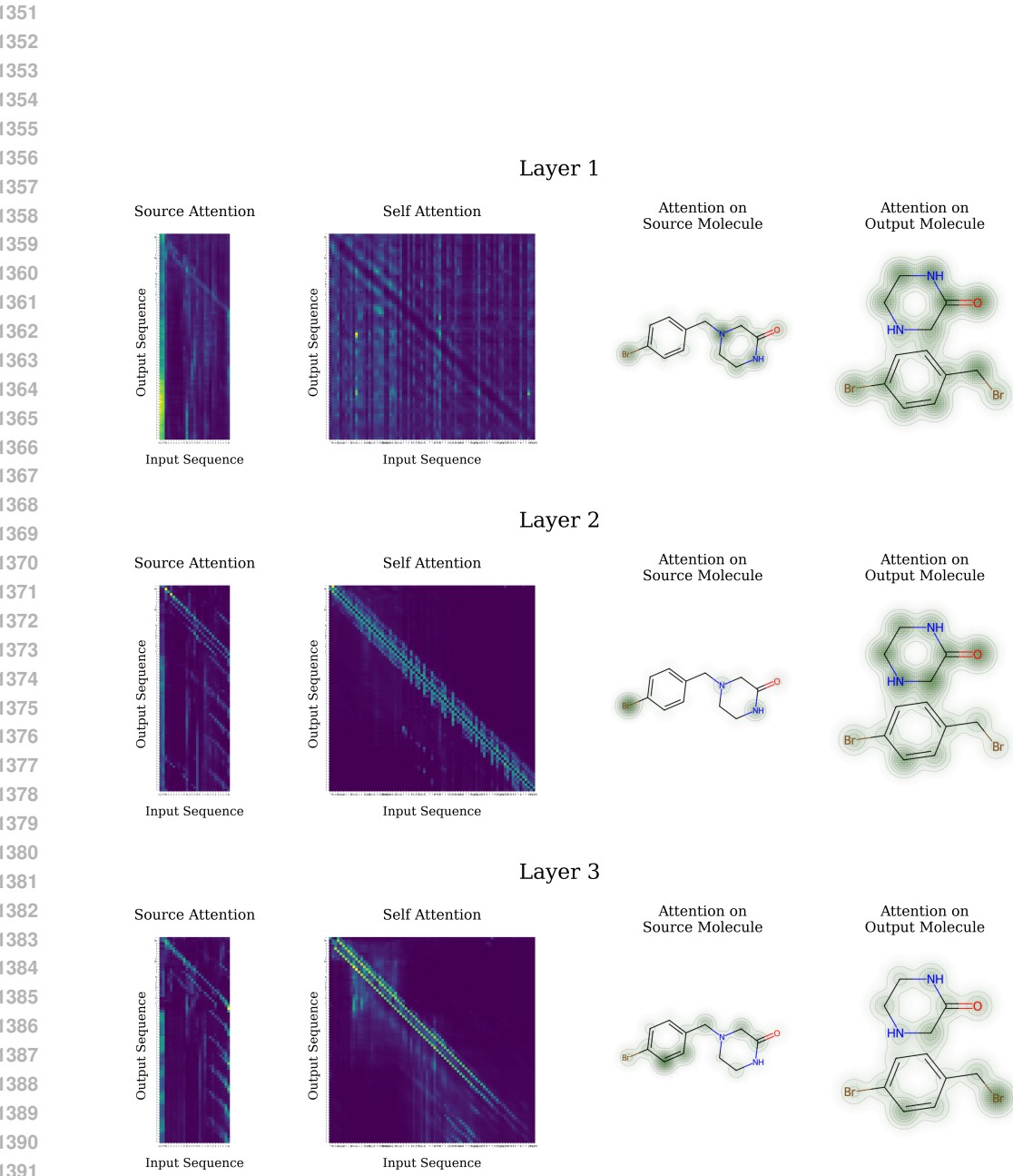

Figure 8: C-DiffER attention maps for layers 1, 2, and 3 on a reaction from the test set halfway through sampling, $t = 0.5$. From left to right: attention on the input product SMILES $y_0$, attention on the noisy sample $x_t$, attention mapped to the input product molecule, and attention mapped to the final predicted output molecule.

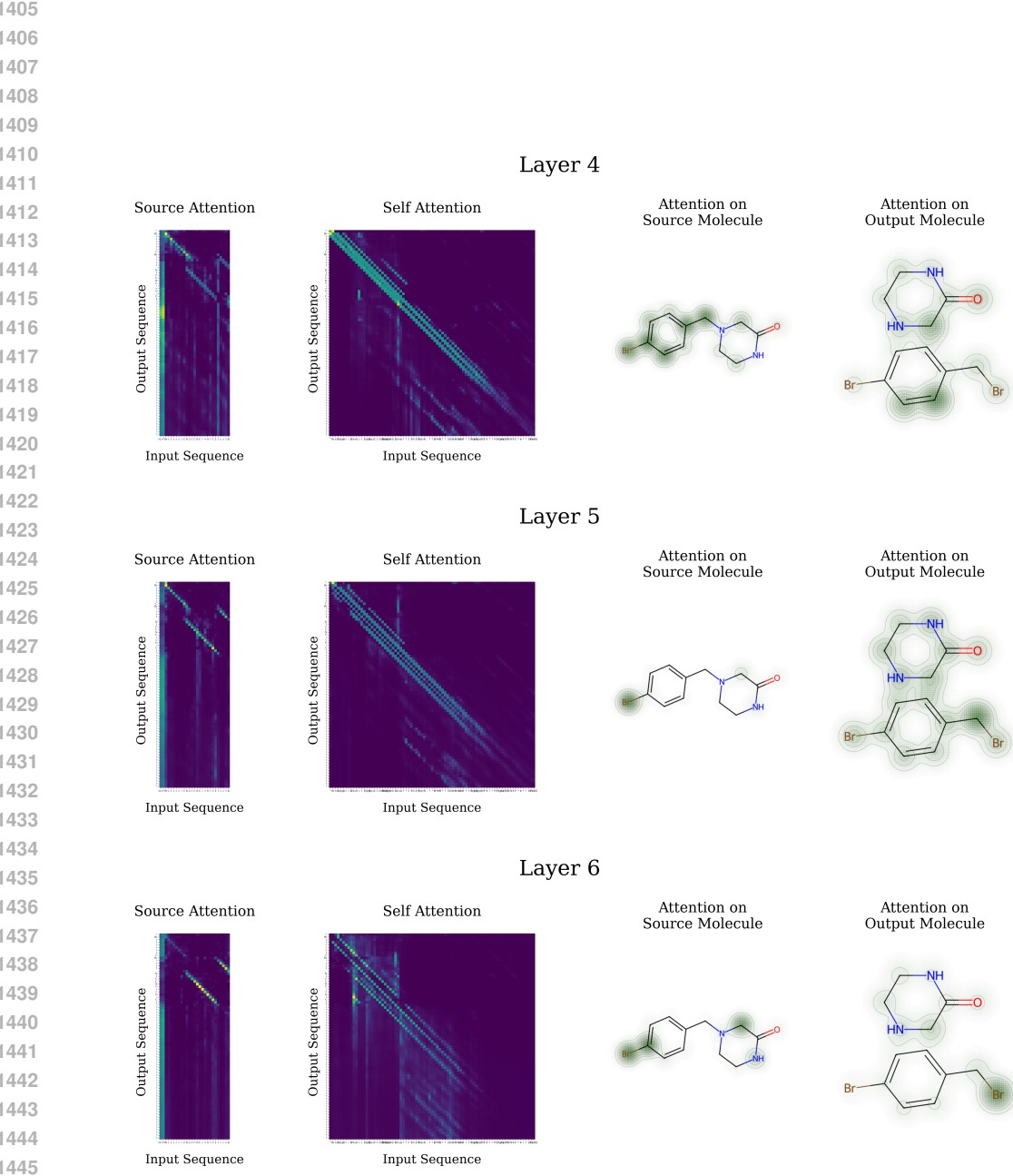

Figure 9: C-DiffER attention maps for layers 4, 5, and 6 on a reaction from the test set halfway through sampling, $t = 0.5$. From left to right: attention on the input product SMILES $y_0$, attention on the noisy sample $x_t$, attention mapped to the input product molecule, and attention mapped to the final predicted output molecule.