# OpenReview forum: "DiffER$^2$: Diffusion Ensembles for Retrosynthesis Prediction with SMILES Adapted Particle Guidance"
_ICLR.cc/2026/Conference — ICLR 2026 Conference Withdrawn Submission_

### Official Review · Reviewer_Ur6v · 2025-10-26

**Soundness:** 2
**Presentation:** 2
**Contribution:** 2
**Rating:** 2
**Confidence:** 5

**Summary:**

The paper builds an ensemble of discrete- and continuous-time diffusion models for molecular generation, augmented with a guidance mechanism to promote output diversity.

**Strengths:**

1. By integrating the strengths of multiple diffusion models, the proposed approach substantially improves performance, while the SMILES-adapted guidance mechanism further enhances sampling diversity—an essential factor for achieving high top-k accuracy in retrosynthesis.

2. The manuscript is clearly written, and the problem formulation is well defined and easy to follow.

**Weaknesses:**

1. The contribution appears incremental: no fundamentally new algorithm is proposed. The method mainly ensembles existing diffusion models and introduces a guidance strategy to refine sampling, which may be viewed as an engineering extension rather than a substantive conceptual advance.

2. Table 1 indicates that the ensemble yields only marginal improvements over the DiffER baseline, casting doubt on the practical effectiveness and necessity of the proposed approach.

3. The paper does not report the training and inference time or computational complexity of using multiple diffusion models, raising concerns that the proposed approach may be computationally expensive in practice.

**Questions:**

N/A

---

### Official Review · Reviewer_37Em · 2025-10-29

**Soundness:** 3
**Presentation:** 2
**Contribution:** 2
**Rating:** 4
**Confidence:** 4

**Summary:**

DiffER2 introduces a unified diffusion ensemble framework for single-step chemical retrosynthesis prediction, extending the categorical diffusion formulation of DiffER by integrating both discrete-time and continuous-time diffusion models. In addition, the work introduces SMILES adapted particle guidance, an inference-time potential that applies a learned repulsive term during sampling to mitigate oversampling of posterior peaks. DiffER2 demonstrates that the continuous-time and discrete-time ensemble, along with particle guidance, yields improvements in both top-k accuracy and the number of unique molecules generated.

**Strengths:**

1. DiffER2 successfully implements SMILES adapted particle guidance to reward intermediate noised SMILES strings that are likely to become distinct molecules. SMILES adapted particle guidance improves diversity relative to the base DiffER model.
2. The authors provide detailed analyses describing the relationship between model performance and various SMILES properties for both DiffER2 and competitive baselines.

**Weaknesses:**

1. The authors' choice to include 4 continuous-time and 4 discrete-time discrete diffusion models in the ensemble is not justified with an ablation study.
2. When reporting the mean and variance of the number of unique reactant generated per reaction in the USPTO test set, it is unclear how many total samples are generated from the ensemble. This context is important to understanding the impact of SMILES adapted particle guidance on diversity.
3. Performance gains relative to the base model DiffER due to particle guidance are rather minor, and DiffER2 remains outperformed in Top-5 and Top-10 accuracy by R-SMILES.
4. Although presented in the Methods section (section 3) along with the novel particle guidance approach, the described length prediction module appears unmodified from the original DiffER manuscript.

Minor:
1. In the "evaluation metrics" section, there is an appendix reference that does not point to a valid appendix section.

**Questions:**

1. Is $f^{\Phi}_{K}$ trained exclusively during sampling? If so, would it be possible to train it during the overall DiffER2 training process?
2. Distinct reactants are not necessarily chemically diverse. Would it be worthwhile to train a version of $f^{\Phi}_{K}$ that is instead a regressor predicting, for instance, Tanimoto distance between two predicted denoised SMILES? Would the increased diversity requirement have a negative or positive effect on Top-N accuracy?

---

### Official Review · Reviewer_ogU1 · 2025-10-31

**Soundness:** 3
**Presentation:** 2
**Contribution:** 2
**Rating:** 4
**Confidence:** 3

**Summary:**

This paper builds upon the previous DiffER model for single-step retrosynthesis prediction. The core task is to generate plausible reactant molecules (as SMILES strings) given a target product molecule. The authors' main contributions are :

1.Ensemble Expansion: They extend the DiffER ensemble to include not only discrete-time diffusion models but also continuous-time discrete diffusion models.

2.Novel Guidance Mechanism: They propose a SMILES-adapted particle guidance method. This is an adaptation of classifier-based guidance designed to increase the diversity of generated molecules by encouraging the model to produce a set of dissimilar outputs during sampling.

3.Empirical Analysis: They provide an in-depth comparison between their diffusion-based approach and auto-regressive models (specifically R-SMILES), analyzing performance relative to various molecular properties (e.g., sequence length change, ring modifications).

**Strengths:**

1.The application of particle guidance to discrete-sequence generation (specifically SMILES strings) is a novel and creative contribution. Translating the concept from continuous spaces (like images) to a discrete, non-differentiable domain is non-trivial. The proposed method of using a classifier to predict molecular dissimilarity from noisy intermediate states is an elegant and pragmatic solution to the problem of defining a kernel on SMILES strings.

2.The hybrid ensemble of discrete-time and continuous-time diffusion models for this task is also a fresh approach. While both types of models have been explored separately, their combination in an ensemble for molecular generation is a thoughtful attempt to leverage their complementary strengths.

3.The methodological foundation is sound and well-grounded. The paper demonstrates a strong command of the relevant literature on both diffusion models and retrosynthesis. The formalization of the unified diffusion framework (following Zhao et al., 2024) and the adaptation of classifier guidance (following Schiff et al., 2024) is technically rigorous.

**Weaknesses:**

1.The central claim of "improving top-k accuracy" is only partially supported. As evident in Table 1, the improvements over the original DiffER and the current state-of-the-art (R-SMILES) are incremental at best, and arguably insignificant in a practical sense. The primary achievement appears to be a redistribution of accuracy: slightly better top-1/top-3 than R-SMILES, but worse top-5/top-10. This suggests the method is better at finding the single best candidate but worse at covering the long tail of plausible candidates compared to the auto-regressive baseline. This contradicts the stated goal of "greater coverage of the posterior."

2.Figure 4 is critical to the authors' argument that their model has unique strengths, but it is confusing and lacks the necessary rigor to support its claims conclusively.

a. Inconsistent X-axes: The subplots for "Length Diff." and "Edit Distance" have completely different scales and meanings on the x-axis. One is a difference in token length, the other is (presumably) a Levenshtein distance. This makes cross-property comparison difficult.

b.Lack of Context: The plots show a "line of best fit," but without showing the underlying data distribution (e.g., as a scatter plot with transparency), it is impossible to judge if the relationship is driven by a few outliers or is consistent across the dataset.

**Questions:**

1. The experimental results in Table 1 show that DiffER2PG+ does not hold a clear, significant advantage over R-SMILES, trading off performance at different k-values. Given the increased complexity of training an ensemble of diffusion models with a separate guidance network, what is the compelling practical reason for a practitioner to adopt DiffER2 over the simpler and highly effective R-SMILES? Could the authors contextualize the trade-off between marginal gains in top-1/3 and the loss in top-10 performance?

2.Could the authors provide scatter plots for Figure 4, showing the underlying data points for both models? This would allow the reader to assess the density of data supporting the regression lines and the presence of any outliers.

---

### Note · Authors · 2025-11-24

**Comment:**

We thank all reviewers for their responses and questions, and we have adjusted future versions of our paper to clarify the analysis presented in the results section and refine the pros and cons of our methodology. These suggestions have helped us further describe the different capabilities between diffusion and auto-regressive models for reactant generation, and we have generated additional results to support our conclusions in future work.

**Withdrawal Confirmation:**

I have read and agree with the venue's withdrawal policy on behalf of myself and my co-authors.